# Phylogenetic Groups and Antimicrobial Resistance Genes in *Escherichia coli* from Different Meat Species

**DOI:** 10.3390/antibiotics10121543

**Published:** 2021-12-16

**Authors:** Angelika Sacher-Pirklbauer, Daniela Klein-Jöbstl, Dmitrij Sofka, Anne-Béatrice Blanc-Potard, Friederike Hilbert

**Affiliations:** 1Institute of Food Safety, Department for Farm Animals and Veterinary Public Health, University of Veterinary Medicine, 1210 Vienna, Austria; Angelika.Sacher-Pirklbauer@wien.gv.at (A.S.-P.); Dimitrij.Sofka@vetmeduni.ac.at (D.S.); 2Section of Herd Management, Clinic for Ruminats, University of Veterinary Medicine, 1210 Vienna, Austria; daniela.klein@vetmeduni.ac.at; 3Laboratory of Pathogen-Host Interactions (LPHI), Université Montpellier, 34095 Montpellier, France; anne.blanc-potard@umontpellier.fr; 4CNRS, UMR 5235, 34095 Montpellier, France

**Keywords:** antibiotic resistance, *Escherichia coli*, foodborne pathogen, phylogroup, poultry meat, beef, pork

## Abstract

*Escherichia coli* isolated from meat of different animal species may harbour antimicrobial resistance genes and may thus be a threat to human health. The objectives of this study were to define antimicrobial resistance genes in *E. coli* isolates from pork, beef, chicken- and turkey meat and analyse whether their resistance genotypes associated with phylogenetic groups or meat species. A total number of 313 *E. coli* samples were isolated using standard cultural techniques. In 98% of resistant isolates, a dedicated resistance gene could be identified by PCR. Resistance genes detected were *tet*(A) and *tet*(B) for tetracycline resistance, *strA* and *aadA*1 for streptomycin resistance, *sul*I and *sul*II for resistance against sulphonamides, *dfr* and *aphA* for kanamycin resistance and *bla*_TEM_ for ampicillin resistance. One *stx1* harbouring *E. coli* isolated from pork harboured the *tet*(A) gene and belonged to phylogenetic group B2, whilst another *stx1* positive isolate from beef was multi-resistant and tested positive for *bla*_TEM,_
*aphA*, *strA–B*, *sul*II, and *tet*(A) and belonged to phylogenetic group A. In conclusion, the distribution of resistance elements was almost identical and statistically indifferent in isolates of different meat species. Phylogenetic groups did not associate with the distribution of resistance genes and a rather low number of diverse resistance genes were detected. Most *E. coli* populations with different resistance genes against one drug often revealed statistically significant different MIC values.

## 1. Introduction

Antimicrobial resistance (AMR) and its spread is one of the most important health problem nowadays. Indicator microorganisms for AMR such as *Escherichia coli* (*E. coli*) are important indicator bacteria for faecal contamination and the occurrence of AMR of animal and human origin. As a commensal and pathogen in warm-blooded animals and humans, *E. coli* can be isolated from different sources such as faeces, manure, water and food of animal and plant origin.

Isolates from food-producing animals and their food pose a direct link from animals to humans and between locations where the primary antibiotic usage takes place (animal and human medicine). Hence, antibiotic-resistant (ABR) isolates are commonly found and may provide insights into upcoming problems but may also provide foreseeable solutions. The analysis of resistant bacteria by whole genome sequencing is becoming a standard procedure but discrimination is very high and one can often “miss the forest for the trees” while waiting for “cut-off values” for epidemiological applications [1,2].

In *E. coli*, which has a rather strong phylogenetic structure, the four major phylogroups (A, B1, B2, and D) can provide a framework. The analysis of these phylogenetic groups can be performed by a simple multiplex PCR [3,4,5] but can also be extracted from sequence data. Phylogroups A and B1 are said to be more generalist, which can associate with multidrug resistance and can be a potential transfer vehicle for AMR between animal species, the environment, and humans [6]. Other studies comparing resistant and non-resistant *E. coli* [7,8] were not able to link specific AMR patterns to certain *E. coli* phylogroups.

Isolates of groups B1 and A can be found in humans, poultry, ruminants, and pigs [9,10,11,12] whereas groups B2 and D are frequently isolated as extraintestinal pathogens from humans and birds [9,13]. The contamination of food of animal origin by *E. coli* may happen during slaughter processing steps for instance skinning, defeathering, or evisceration. *E. coli* isolated from meat may harbour antimicrobial resistance genes but may also be pathogenic to humans for instance STEC and thus be a threat to human health [14,15].

To detect whether the distribution of phylogenetic groups in resistant and non-resistant *E. coli* differs between most commonly used meat species, we analysed antimicrobial-resistant, -sensitive, and -intermediate *E. coli* isolated from chicken meat, pork, beef, and turkey meat for antimicrobial resistance genes and defined their phylogenetic group.

## 2. Results

In this study, we tested a total number of 313 *E. coli* isolates from different meat species for resistance to most important veterinary drugs in use and defined resistance genes and phylogenetic groups.

### 2.1. Phenotypical Resistance

Most of the resistant isolates were resistant to tetracycline (*n* = 230), followed by sulphonamides (*n* = 142), ampicillin (*n* = 89), trimethoprim (*n* = 74), streptomycin (*n* = 50), and kanamycin (*n* = 28) by using the disc diffusion assay and the microbroth dilution assay. Fifty-nine isolates exhibited an intermediate phenotype against streptomycin by using the disc diffusion test. Three isolates were intermediate resistant to ampicillin, and one isolate to sulphonamide by using the microbroth dilution assay.

### 2.2. Tetracycline Resistance Genes

We tested for several genes known to cause resistance against tetracycline in *E. coli* as are *tet*(A), *tet*(B), *tet*(C), *tet*(G). Only two of these known resistance genes were detected in our isolates, namely *tet*(A) and *tet*(B). All resistant isolates harboured at least one resistance determinant. The *tet*(A) gene fragment (*n* = 156) was more frequently detected than the *tet*(B) gene fragment (*n* = 76). Two of these isolates (one pork and one beef isolate) harboured both genes. Twenty PCR fragments were sequenced to confirm the partial tetracycline resistance gene sequence. No sequence variations that would have caused an amino acid sequence change were found within these sequences. Comparing the minimum inhibitory tetracycline concentration of the isolates and the detection of resistance genes, a significant difference (*p* < 0.05) was found within the distribution (Figure 1).

Higher minimum inhibitory concentrations were found for isolates harbouring the *tet*(B) gene fragment. No differences were found between resistance gene distribution and meat species (Figure 2), whereas distribution between *tet*(A) and *tet*(B) in minced meat (mixed pork and beef)—with *tet*(B) as the predominant gene—was significantly different to the meat of the single meat species (*p* < 0.05).

### 2.3. Sulphonamide Resistance Genes

We tested the occurrence of *sul*I and *sul*II resistance gene fragments. The *sul*II gene fragment was found more often (*n* = 113) than the *sul*I gene fragment (*n* = 32) in a total of 142 sulphonamide-resistant *E. coli* isolates. Three isolates harboured both gene fragments (one pork, one chicken meat, and one minced meat isolate). Sequence analysis was performed in a random sample of *sul*I (*n* = 10) and *sul*II (*n* = 10) positive isolates and on the only intermediate-resistant isolate (*sul*II). No differences were detected in the coding sequence of the genes.

### 2.4. Streptomycin Resistance Genes

Streptomycin resistance in *E. coli* is recommended to be only tested as a disc diffusion assay [16]. For streptomycin resistance, many isolates were determined as intermediate resistant. Hence, resistance gene detection was performed for all intermediate-resistant isolates as well. In most resistant isolates (*n* = 50, 94%) the *strA*–*strB* gene cluster could be detected by PCR of the *strA* gene fragment. In three resistant isolates, the *aadA1* gene fragment was amplified (6%). In all intermediate isolates, a resistance gene fragment could be detected by PCR (the *strA*–*strB* in 40 isolates, the *aadA1* in 22 and both genes in three isolates). The *strA*–*strB* cluster was associated with higher inhibitory concentrations, as shown in the histogram of the disc diffusion assay (Figure 3).

### 2.5. Ampicillin Resistance Genes

In this study, we used primers to detect the coding region for different β-lactamases: Temoniera (TEM), cephalomycinase (CMY), Pseudomonas-specific-enzyme group (PSE), active on imipenem (IMP), oxacillinase (OXA). By using distinct primers, we were able to detect the *bla*_TEM_ gene fragment in all of our resistant isolates. One intermediate isolates harboured the *bla*_TEM_ gene fragment and another intermediate isolate harboured the *bla*_OXA_ gene fragment. No resistance gene was found in the third intermediate isolate.

### 2.6. Trimethoprim Resistance Genes

In 71 isolates, a *dfr*A, and in 3 isolates (two chicken meat isolates, one minced meat isolate), a *dfr*7&17 gene fragment could be amplified. In four trimethoprim-resistant isolates (one pork isolate and three chicken meat isolates), none of the tested resistance gene fragments was detected.

### 2.7. Kanamycin Resistanc Genes

In all 28 resistant isolates, the *aphA*-1A resistance gene fragment was amplified. Isolates originated from pork (*n* = 6), beef (*n* = 5), chicken meat (*n* = 14), turkey meat (*n* = 1), and minced meat (*n* = 2).

### 2.8. Multi-Resistant Isolates and Gene Association

From all isolates, harbouring resistance elements (phenotypically resistant and intermediate isolates), only 75 carried only one resistance determinant (single resistance) but all the rest of the isolates harboured at least two or more resistance gene elements (*n* = 190). In more than 82% (*n* = 61) of the isolates carrying a single resistance gene, this gene was a tetracycline resistance gene—either *tet*(A) in 46 isolates and *tet*(B) in 15 isolates. Isolates with multiple resistance gene fragments were found in isolates from all meat species. Resistance genes associated with other resistance genes are stated in Table 1. Most often associated the trimethoprim resistance gene *dfr*A, the *strA–B* cluster and the *sul*II genes. They were associated with each other or with other’s resistance genes. Transformation studies confirmed these results as a combined transfer of pheno- and genotypically resistance could be detected even though this was not the case for every transformant. This has been shown for three plasmids all isolated from three different multidrug-resistant isolates from this study, which were individually transformed into a DH5α-competent strain. Transformants exhibited phenotypical combined resistance to tetracycline, sulphonamides, and streptomycin, combined resistance to tetracycline and streptomycin and single tetracycline resistance. Five other isolated plasmids were not harbouring the expected genes or were simply not transformable into DH5α. In none of the sensitive tested isolates was a resistance gene identified.

### 2.9. Phylogenetic Groups

Phylogenetic group A was the most common group among isolates from all meat sources, varying between 73% in minced meat and 56% in poultry meat. Phylogroup B1 was the second most common group with 28% in beef and poultry, 27% in minced meat, 24% in pork and 17% in turkey meat. Less commonly found were phylogroup D with 0–17% and B2 with only 0–6%. These differences are significant within the meat source (*p* < 0.05). There were no significant association between phylogenetic groups and the different resistance genes.

## 3. Discussion

As antimicrobial resistance is a global fear to public health, many industrialised as well as developing countries have established monitoring programs. The main problems in achieving comparable data are, for instance, the different methods used to detect antimicrobial resistance. Genetic methods for the detection of antimicrobial resistance genes appear to be the most discussed approach for overcoming this problem [1,2]. Despite the fact that there are many studies on the dissemination and incidence of resistance genes in diverse microorganisms nowadays, new data are still necessary [9,15,17,18]. In this study, we detected the appropriate resistance gene fragment in 257 out of 261 phenotypically resistant and in 62 of 63 isolates with an intermediate resistance phenotype by PCR—which in both cases is more than 98%. In spite of the large number of known resistance genes, for instance, for tetracycline and ampicillin resistance, in our isolates, only certain genes were commonly found (*tet*(A) and *tet*(B); *bla*_TEM_). This was rather unexpected as trimmed meat of different animal species may harbour *E. coli* isolates of animal and of human origin [19,20,21].

Antimicrobial resistance profiles of microorganisms isolated from food of animal origin differ according to the animal species [14]. Therefore, we also expected a diversity in the occurrence of resistance genes in isolates of meat of different animal species. However, the distribution of resistance genes did not differ significantly (Figure 2) for all resistance genes detected. This is even more surprising because it is well known that different resistance genes associate and assign with phenotypically dissimilar minimum inhibitory concentrations of the isolate [21,22,23]. In our study, this difference was statistically significant (*p* < 0.05) between *tet*(A) and *tet*(B) as well as between *aadA*1 and *str*A–B harbouring isolates (Figure 1 and Figure 3). No statistically significant divergence could be detected between isolates harbouring either *sul*I or *sul*II genes.

Since isolates carrying multiple resistance genes were detected, research has focused on the combined horizontal transfer of resistance genes within or even across bacterial species. Hence, associations between resistance genes are of special interest, as is if and how horizontal transfer appears. In our study, multi-resistant isolates were detected quite often—a single resistance genotype was detected in only 28% of the tested isolates. This single resistance genotype was most often a *tet*(A) harbouring strain (61%). In all but one intermediate isolates (*n* = 63) could a responsible resistance gene fragment be detected. The only intermediate resistant isolate in which no resistant gene fragment could be detected was a pork *E. coli* isolate with intermediate resistance to ampicillin (MIC of 16 µg/mL). As decreased sensitivity to β-lactam antibiotics has been long known to also be due to changes in the membrane structure of *E. coli* [24], this might be due to this strain as well. These results were confirmed by sequencing a random sample and subsequent alignment. In some of the isolates carrying multiple resistance genes, we could show that isolated plasmids could transfer resistance to the antimicrobial sensitive lab strain DH5α. This was also feasible for combined resistance to tetracycline, sulphonamides, and streptomycin. All the resistant isolates were phylogenetically characterised by phylogenetic grouping. Phylogenetic group A was the most common group among isolates from all meat sources. This varied between 73% in minced meat and 56% in poultry meat. Phylogroup B1 was the second most common group with 28% in beef and poultry, 27% in minced meat, 24% in pork, and 17% in turkey meat. Less commonly found was phylogroup D with 0–17% and B2 with only 0–6%. These differences are significant within the meat source (*p* < 0.05). There were no significant association between phylogenetic groups and the different resistance genes.

Antimicrobial therapy in farm animals is under discussion as each application inevitably leads to resistance development and the horizontal transfer of resistance genes. Despite the potential involvement of human and animal bacterial pathogens, the transfer of resistance within commensals is important as a reservoir of resistance genes. Even though antimicrobial therapy for animal welfare reasons is required in serious bacterial infections, the prudent use of antimicrobial drugs is necessary.

Food of animal origin may harbour antimicrobial-resistant bacteria and thus the education of consumers in regard to handling, meal preparation and consumption must be a priority in public health.

Our study gives essential information to consider new tools for antimicrobial resistance surveillance using sequence-based methods but the limited number of antimicrobials used to test for resistance in our study comprises a weakness and additional studies are required before solely the surveillance of antimicrobial resistance can be performed using genetic tools. Nevertheless, important antimicrobials used in animals have been tested and we did not consider essential antimicrobials used in humans, even though shigatoxin-producing isolates are clearly pathogens, antimicrobial treatment in humans is not recommended as most antimicrobials trigger toxin production, and thus induce severe disease symptoms—whereas other human infections caused by *E. coli* are generally not of orally origin.

In conclusion, our results showed a high occurrence of resistant *E. coli* isolated from poultry meat, pork, and beef; however, a rather low number of diverse resistance genes were detected and they were similarly distributed across all meat species. *E. coli* populations most often exhibited significantly statistically different MIC values when harbouring different resistance genes.

## 4. Materials and Methods

Bacterial strains and sampling procedure: 4 intermediate-resistant, 261 resistant, and 48 sensitive *E. coli* isolates originating from raw meat samples including minced meat were randomly selected and collected throughout Austria’s supermarkets, EU-approved slaughterhouses, butchers, and street markets. *E. coli* isolates were isolated according to ISO 16649-1. From each plate, between two and three isolates were selected for biochemical analysis and identification and for resistance testing. Only one isolate from each sample was included in the study unless the isolates showed a distinct resistance phenotype. Resistant isolates from pork (102 isolates), beef (41 isolates), chicken meat (82 isolates), turkey meat (23 isolates), 13 from minced meat (pork and beef), 4 intermediate, and 48 sensitive isolates were tested for resistance genes using PCR methods (Table 2).

Biochemical analysis of strains: for the biochemical analysis and identification of *E. coli,* the API 20E (BioMerieux 20100, bioMérieux Austria GmbH, Vienna, Austria) was used.

Definition of phylogenetic groups: the method described by Clermont et al., 2000 [3], was used to identify four main phylogenetic groups: A, B1, B2, and D. From all isolates, genomic DNA was extracted from bacterial suspensions using standard techniques [25]. For the purpose of multiplex PCR, the KAPA2G Fast Mulitplex Ready Mix (PEQLAB Biotechnologie GMBH, Erlangen, Germany) was used.

Susceptibility testing: isolates that had not been previously phenotypically analysed were tested using the disc diffusion assay and the microbroth dilution assay, as recommended by the Clinical Laboratory Standards Institute with the control strain ATCC 25922 [16]. Resistance was analysed against most important antimicrobial veterinary drug groups as are ampicillin, kanamycin, streptomycin, sulphonamides, tetracycline, and trimethoprim.

PCR of resistance genes was done using primers and conditions stated in Table 3. 

The sequencing of PCR products: the sequence was determined using the Big Dye Terminator v. 3.1 cycle sequencing kit and an Applied Biosystems 310 ABI Prism Genetic Analyser.

Transformation studies: on selected isolates with single and multiple detected resistance genes, plasmid isolation was performed using a spin mini preparation tool kit (QIAprep Spin Miniprep Kit, Qiagen, Hilden, Germany). From these plasmids, the PCRs of respective resistance genes were carried out. PCR-positive plasmids were chemically transformed into DH5α using standard techniques [25].

Statistical analysis: statistical analyses were carried out using SPSS (Statistical Package for Social Sciences Version 17.0 SPSS Inc., Somers, NY, USA). To examine differences between isolates of different origins, the occurrence of antimicrobial resistance genes and phenotype, *χ*^2^-, and univariable logistic regression tests were used.

## Figures and Tables

**Figure 1 antibiotics-10-01543-f001:**
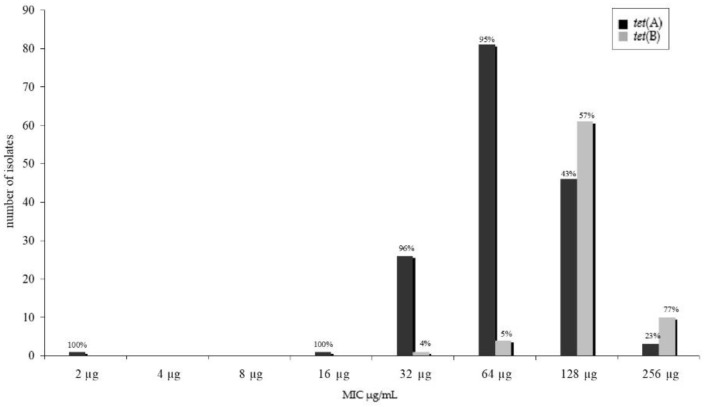
Distribution of *tet*(A) and *tet*(B) gene fragments and resistance phenotype—determined by minimum inhibitory concentration using the microbroth dilution.

**Figure 2 antibiotics-10-01543-f002:**
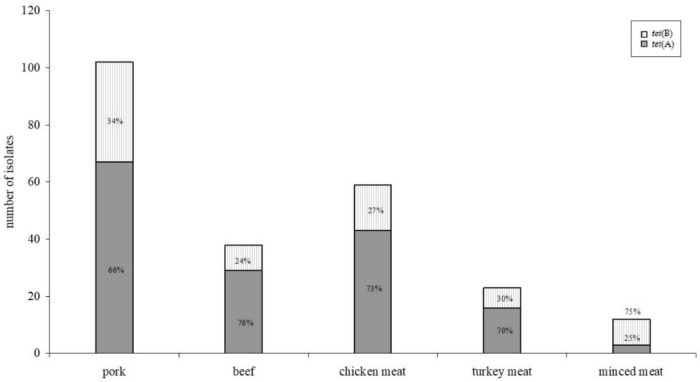
Distribution of *tet*(A) and *tet*(B) gene fragments within different meat species. Despite differences regarding the incidence of resistant isolates in different meat species, no statistically significant (*p* < 0.05) difference could be detected within the distribution of resistance genes.

**Figure 3 antibiotics-10-01543-f003:**
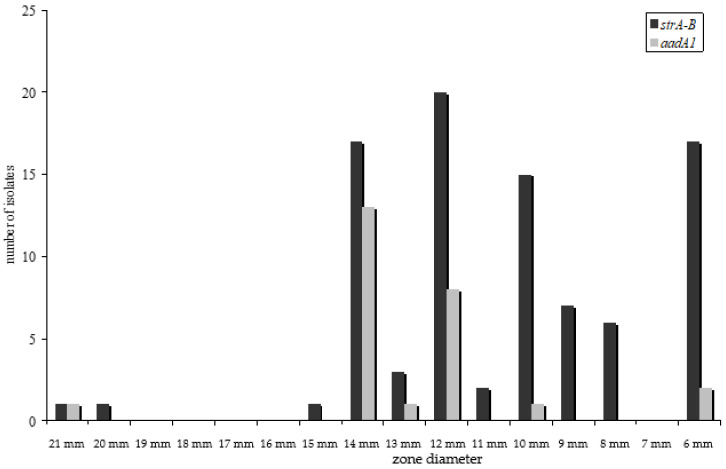
Distribution of *strA–B* and *aadA*1 gene fragments and the resistance phenotype—determined by disc susceptibility testing given by zone diameter. The isolates harbouring the *strA–B* gene cluster developed higher MIC_90_.

**Table 1 antibiotics-10-01543-t001:** Resistance gene association in *E. coli* isolates from meat.

	*tet*(A) *n* = 158	*tet*(B) *n* = 76	*sul*I *n* = 33	*sul*II *n* = 114	*bla*_TEM_ *n* = 93	*dfrA* *n* = 71	*dfr7&17* *n* = 3	*aphA1* *n* = 28	*strA–B* *n* = 89	*addA1* *n* = 26
*tet*(A)	-	2	17	55	47	43	2	13	54	10
*tet*(B)		-	10	37	21	16	1	13	19	8
*sul*I			-	3	12	10	0	5	4	3
*sul*II				-	53	58	3	19	65	4
*bla* _TEM_					-	42	3	17	41	7
*dfrA*						-	0	9	42	3
*dfr7&17*							-	1	1	0
*aphA1*								-	17	2
*strA–B*									-	3
*aadA1*										-

**Table 2 antibiotics-10-01543-t002:** *E. coli* isolates from meat and phylogenetic groups.

Meat Species	Group A	Group B1	Group B2	Group D	Number
Pork	86	29	4	3	122
Chicken meat	56	28	6	9	99
Beef	35	15	1	2	53
Turkey meat	16	4	0	4	24
Minced meat	11	4	0	0	15

**Table 3 antibiotics-10-01543-t003:** Primer and conditions for PCR of tested resistance genes.

Primer Name	Sequence	Target Gene	Amplicon Size	Annealing Temp (°C)	Reference
tet(A)-f tet(A)-r	5′-gctacatcctgcttgtgccttc-3′ 5′-catagatcgccgtgaagagg-3′	*tet*(A)	210	57	[26]
tet(B)-f tet(B)-r	5′-ttggttcggggcaagttttg-3′ 5′-gtaatgggccaataacaccg-3′	*tet*(B)	659	57	[26]
tet(C)-f tet(C)-r	5′-cttgagagccttcaacccag-3′ 5′-atggtcgtcatctacctgcc-3′	*tet*(C)	418	55	[26]
tet(G)-f tet(G)-r	5′-agcaggtcgctggacactat-3′ 5′-cgcggtgttccactgaaaac-3′	*tet*(G)	623	55	[27]
sulI-f sulI-r	5′-tggtgacggtgttcggcattc3′ 5′-gcgaaggtttccgagaaggtg-3′	*sul*I	790	63	[28]
sulII-f sulII-r	5′-gcgctcaaggcagatggcatt-3′ 5′-gcgtttgataccggcacccgt-3′	*sul*II	293	60	[29]
strA–B-f strA_B-r	5′-ccaatcgcagatagaaggcaag-3′ 5′-atcaactggcaggaggaacagg-3′	*strA*	580	65	[30]
aadA1-f aadA1-r	5′-aacgaccttttggaaacttcgg -3′ 5′-ttcgctcatcgccagcccag-3′	*aadA*1	352	60	[30]
bla_CMY_-f bla_CMY_-r	5′-tggccgttgccgttatctac-3′ 5′-cccgttttatgcacccatga-3′	*bla* _CMY_	870	55	[27]
bla_IMP_-f bla_IMP_-r	5′-gaatagagtggattaattctc-3′ 5′-ggtttaayaaaacaaccacc-3′	*bla* _IMP_	232	55	[31]
bla_OXA-2_-f bla_OXA-2_-r	5′-caagccaaaggcacgatagttg-3′ 5′-ctcaacccatcctacccacc-3′	*bla* _OXA_	561	56	[31]
bla_PSE_-f bla_PSE_-r	5′-tgcttcgcaactatgactac-3′ 5′-agcctgtgtttgagctagat-3′	*bla* _PSE_	438	55	[27]
bla_TEM-1_-f bla_TEM-1_-r	5′-cagcggtaagatccttgaga-3′ 5′-actccccgtcgtgtagataa-3′	*bla* _TEM_	643	55	[27]
dfr1-f dfr1-r	5′-gtgaaactatcactaatgg-3′ 5′-ttaacccttttgccagattt-3′	*dfrA1, dfrA5, dfrA15, dfrA15b, dfrA16, dfrA16b*	474	55	[32]
dfr2-f dfr2-r	5′-gatcgcgtgcgcaagaaatc-3′ 5′-aagcgcagccacaggataaat-3′	*dfrB1, dfrB2, dfrB3*	141	60	[32]
dfr7&17-f dfr7&17-r	5′-acatttgactctatgggtgttcttc-3′ 5′-aaaactgttcaaaaaccaaattgaa-3′	*dfr*7&17	280	55	[33]
aphA-1a-f aphA-1a-r	5′-aacgtcttgctcgaggccgcg-3′ 5′-ggcaagatcctggtatcggtctgc-3′	*aphA*-1a	670	65	[34]

## Data Availability

The data presented in this study are available on request from the corresponding author. Requests for isolates/plasmids can be send to the corresponding author.

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
