# Peer review of "Phylogenetic Groups and Antimicrobial Resistance Genes in Escherichia coli from Different Meat Species"

_antibiotics, 2021, doi:10.3390/antibiotics10121543_

Round 1

Reviewer 1 Report

Sacher-Pirkelbauer and co-workers presented the results of a scientific study conducted on bacteria of the genus Escherichia coli isolated from pork, beef, turkey and chicken. The focus of the study was to examine the presence of a phylogenetic group and antibiotic resistance genes: ampicillin, kanamycin, streptomycin, sulfonamides, tetracycline, and trimethoprim in isolated E.coli strains.

The problem of antimicrobial resistance is ubiquitous. With all the measures implemented to reduce antimicrobial resistance, it is very important, even fateful, to investigate and quantify existing food "contamination" of antibiotics and the presence of resistance genes in them because it depends on the future effectiveness of antibiotics in the not so distant future. This is why the paper has a very important place in determining further directions for solving this problem.

The work is very well conceived, several methods were used that I consider appropriate, they are not described in detail, but in the disk diffusion and microdilution test the standard according to which the testing was performed is stated.

Likewise, there is no detailed description of molecular methods but the paper contains references which show in detail described PCR reactions and electrophoresis conditions.

The results in the tables and graphs are clearly shown.

Since 30% of the literature citations have been published within the last 5 years, I find this satisfactory.

It is necessary to correct:

Line 25: Please Escherichia coli write in Italic

Line 37: Please define ABR

Line 124: Please define TEM, CMY, PSE, IMP, OXA.

Line 205: Please E.coli write in Italic

Author Response

Dear reviewer, 

Many thanks for your review and corrections on the manuscript antibiotics-1466629. We corrected as follows: 

Line 25: Escherichia coli written in Italic

Line 37: ABR defined as antibiotic resistant

Line 124: TEM defined as Temoniera, CMY defined as cephalomycinase, PSE defined as "Pseudomonas-specific-enzyme group", IMP defined as "active on imipenem", OXA defined as oxacillinase.

Line 205: E. coli written in Italic

Best regards Friederike Hilbert

Reviewer 2 Report

This study shows phylogenetic groups and antimicrobial resistance genes in Escherichia coli from different meat species. None of innovative results presented in the current manuscript. It is known that the phylogroups of Escherichia coli from different host sources show a clear difference. And this study just introduces the drug resistance of several common antibiotics and the distribution of drug resistance genes in E. coli isolates. The data presented in the current article are not sufficient to be published in Antibiotics. In addition, we also proposed the several opinions and concerns, as follows.

1. The total number of strains used in the study should be given in the abstract. Similarly, the total number of strains used should be given at the beginning of the results.
2. Please give the reference or basis of the bacterial resistance breakpoint standard in the method .
3. It is recommended that more drugs should be tested in drug sensitivity experiments, especially those used commonly in hospitals. This is very important and meaningful data.
4. Line 86-88 : (Figure 1 legend), it should not be written in the legend. It should be written in the discussion.
5. Line 83, 92 : “(see Figure 1)” , the word “See” should be deleted. Please check the entire manuscript.
6. Line 108: please indicate the references.
7. Line 146-148: Did you do "Transformation studies" to verify this sentence? If not, please cite the reference. If yes, please add relevant experimental methods and results.
8. “2.9. Phylogenetic groups”: It is suggested that a picture should be added to show the result clearly.
9. Line 155-156: Please rewrite“Less commonly found were phylogroup D with 0 to 17% and B2 with only 0 to 6%. ”Please indicate which strains does 0, 17% and 6% refer to?
10. Line 163, 170: Do not use "e.g." , because this is not a normal word.
11. Line184, 186: “multiresistant isolates” should change to “multidrug-resistant”. However, multidrug-resistant bacteria refer to resistance to three or more antimicrobial drugs, and in this study it should be written as "carrying multiple resistant genes", otherwise it will cause misunderstanding.
12. The significance of the main findings should be discussed deeply, for example, what kind of guidance does it have for the work of the farm? What are the requirements for supermarket quarantine work?
13. Strengths and Limitation of Research is lacking, please add strength and limitation of your manuscript. 
14. Please add a paragraph of conclusion to illustrate the importance of research.

Author Response

Dear reviewer,

We like to thank the reviewer for valuable suggestions given and changed as follows:

  1. Total number of isolates (strains) are now given in the abstract and the total number of isolates (strains) used is given at the beginning of the results section.
  2. Reference of the bacterial resistance breakpoint standard is given in the method section already. It is CLSI standards and also break points given there.
  3.  Only most important antimicrobials used in animals have been tested. For the STX positive isolates, clearly pathogens, antimicrobial treatment in humans with infections caused by STX producing E. coli is not recommended but even opposed as most antimicrobials trigger toxin production and thus induce severe disease symptoms. Whereas other human infections caused by E. coli are generally not caused by meat isolates. This explanation is now also part of the revised discussion section. 
  4. To highlight results shown in figure 1,2 and 3  the discussion section was modified.
  5. The word “See” has been exchanged throughout the document.
  6. Appropriate reference is given now. (CSLI standards) 
  7. Transformation studies have been added in the m&m and results section
  8. 2.9. Phylogenetic groups”: An additional table was added with all data
  9. Whereever e.g. has been used, it has been exchanged 
  10. Line184, 186: “multiresistant isolates” has been change to "carrying multiple resistant genes"
  11. Discussion was changed regarding the suggestions of the reviewer
  12. “Strengths and Limitation” has been added to the discussion section of the manuscript
  13. We included a conclusion section

Additionally, we disagree with the reviewer’s statement that no innovative results are presented in the current manuscript. But we agree that these sections had to be highlighted in the manuscript. As this is the first study to our knowledge that shows equal distribution of tetracycline genes in meat of different meat species despite the difference in the occurrence of tetracycline resistant isolates. Also studies on quantitative difference in MIC in combination with different genes are scarce. Although it is common knowledge that different resistance genes can develop different phenotypic resistance. 

Best regards Friederike Hilbert in the name of all authors

Reviewer 3 Report

General: The research is dedicated in the phylogenetic group and antimicrobial resistance genes in E. coli isolates from pork, beef, chicken- and turkey meat.  The results are important and the application in food safety looks visible.

Title: Phylogenetic groups and antimicrobial resistance genes in  Escherichia coli from different meat species 

Abstract: Lacks to mention specific objective of the work, and materials and methods. It needs also a conclusive  statement.

Introduction : Line 37...Hence, antibiotic resistance residue  (ABR) isolates are commonly ...... mention what ABR abbreviate.

Materials and methods: Sample is collected from pork, beef, chicken- and turkey meat.  information missed are:- a) there is no information on the nature of the sample is that ready to eat (RTE), partially processed, raw?  b) mention how the E.coli is isolated from the food matrices.

It will be advisable to mention conclusive remarks after the discussion.

Author Response

Dear reviewer,

Many thanks for suggestions on the manuscript antibiotics-1466629. We changed the manuscript according your suggestions as follows.

Abstract: Specific objectives were added, a short m&m section was included and the abstract ends now with a conclusive  statement.

Introduction :  In Line 37...ABR has been defined as antibiotic resistance. 

Materials and methods

For a) it was added in the text as follows: E. coli isolates originated from raw meat samples, including minced meat randomly selected and collected throughout Austria’s supermarkets, EU-approved slaughterhouses, butchers, and street markets.

For b) we added in the text: E. coli were isolated according to ISO 16649-1. From each plate two to three isolates were selected for biochemical analysis and identification and for resistance testing. Only one isolate from each sample was included in the study unless isolates showed a distinct resistance phenotype.

A concluding section was added.

Best regards Friederike Hilbert in the name of all authors

Round 2

Reviewer 2 Report

I have no comments for authors now.